# Scaling Laws for a Multi-Agent Reinforcement Learning Model

**Oren Neumann & Claudius Gros**
Institute for Theoretical Physics
Goethe University Frankfurt
Frankfurt am Main, Germany
`{neumann,gros}@itp.uni-frankfurt.de`

## Abstract

The recent observation of neural power-law scaling relations has made a significant impact in the field of deep learning. A substantial amount of attention has been dedicated as a consequence to the description of scaling laws, although mostly for supervised learning and only to a reduced extent for reinforcement learning frameworks. In this paper we present an extensive study of performance scaling for a cornerstone reinforcement learning algorithm, AlphaZero. On the basis of a relationship between Elo rating, playing strength and power-law scaling, we train AlphaZero agents on the games Connect Four and Pentago and analyze their performance. We find that player strength scales as a power law in neural network parameter count when not bottlenecked by available compute, and as a power of compute when training optimally sized agents. We observe nearly identical scaling exponents for both games. Combining the two observed scaling laws we obtain a power law relating optimal size to compute similar to the ones observed for language models. We find that the predicted scaling of optimal neural network size fits our data for both games. We also show that large AlphaZero models are more sample efficient, performing better than smaller models with the same amount of training data.

## 1 Introduction

In recent years, power-law scaling of performance indicators has been observed in a range of machine-learning architectures (Hestness et al., 2017; Kaplan et al., 2020; Henighan et al., 2020; Gordon et al., 2021; Hernandez et al., 2021; Zhai et al., 2022), such as Transformers, LSTMs, Routing Networks (Clark et al., 2022) and ResNets (Bello et al., 2021). The range of fields investigated include natural language processing and computer vision (Rosenfeld et al., 2019). Most of these scaling laws regard the dependency of test loss on either dataset size, number of neural network parameters, or training compute. The robustness of the observed scaling laws across many orders of magnitude led to the creation of large models, with parameters numbering in the tens and hundreds of billions (Brown et al., 2020; Hoffmann et al., 2022; Alayrac et al., 2022).

Until now, evidence for power-law scaling has come in most part from supervised learning methods. Considerably less effort has been dedicated to the scaling of reinforcement learning algorithms, such as performance scaling with model size (Reed et al., 2022; Lee et al., 2022). At times, scaling laws remained unnoticed, given that they show up not as power laws, but as log-linear relations when Elo scores are taken as the performance measure in multi-agent reinforcement learning (MARL) (Jones, 2021; Liu et al., 2021) (see Section 3.2). Of particular interest in this context is the AlphaZero family of models, AlphaGo Zero (Silver et al., 2017b), AlphaZero (Silver et al., 2017a), and MuZero (Schrittwieser et al., 2020), which achieved state-of-the-art performance on several board games without access to human gameplay datasets by applying a tree search guided by a neural network.

Here we present an extensive study of power-law scaling in the context of two-player open-information games. Our study constitutes, to our knowledge, the first investigation of power-law scaling phenomena for a MARL algorithm. Measuring the performance of the AlphaZero algorithm using Elo rating, we follow a similar path as Kaplan et al. (2020) by providing evidence of power-law

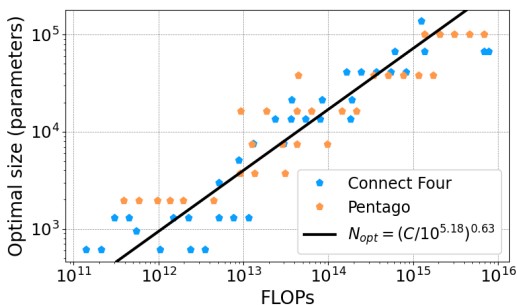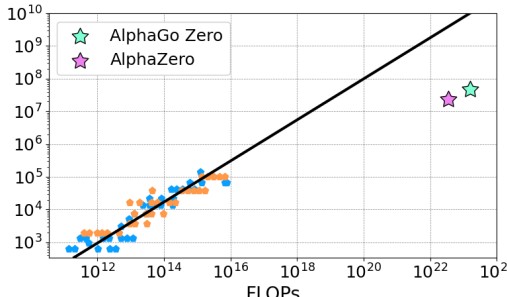

Figure 1: **Left:** Optimal number of neural network parameters for different amounts of available compute. The optimal agent size scales for both Connect Four and Pentago as a single power law with available compute. The predicted slope $\alpha_C^{opt} = \alpha_C/\alpha_N$ of Eq. (7) matches the observed data, where $\alpha_C$ and $\alpha_N$ are the compute and model size scaling exponents, respectively. See Table 3 for the numerical values. **Right:** The same graph zoomed out to include the resources used to create AlphaZero (Silver et al., 2017a) and AlphaGoZero (Silver et al., 2017b). These models stand well bellow the optimal trend for Connect Four and Pentago.

scaling of playing strength with model size and compute, as well as a power law of optimal model size with respect to available compute. Focusing on AlphaZero-agents that are guided by neural nets with fully connected layers, we test our hypothesis on two popular board games: Connect Four and Pentago. These games are selected for being different from each other with respect to branching factors and game lengths.

Using the Bradley-Terry model definition of playing strength (Bradley & Terry, 1952), we start by showing that playing strength scales as a power law with neural network size when models are trained until convergence in the limit of abundant compute. We find that agents trained on Connect Four and Pentago scale with similar exponents.

In a second step we investigate the trade-off between model size and compute. Similar to scaling observed in the game Hex (Jones, 2021), we observe power-law scaling when compute is limited, again with similar exponents for Connect Four and Pentago. Finally we utilize these two scaling laws to find a scaling law for the optimal model size given the amount of compute available, as shown in Fig. 1. We find that the optimal neural network size scales as a power law with compute, with an exponent that can be derived from the individual size-scaling and compute-scaling exponents. All code and data used in our experiments are available online [1]

## 2 RELATED WORK

Little work on power-law scaling has been published for MARL algorithms. Schrittwieser et al. (2021) report reward scaling as a power law with data frames when training a data efficient variant of MuZero. Jones (2021), the closest work to our own, shows evidence of power-law scaling of performance with compute, by measuring the performance of AlphaZero agents on small-board variants of the game of Hex. For board-sizes 3-9, log-scaling of Elo rating with compute is found when plotting the maximal scores reached among training runs. Without making an explicit connection to power-law scaling, the results reported by Jones (2021) can be characterized by a compute exponent of $\alpha_C \approx 1.3$, which can be shown when using Eq. 3. In the paper, the author suggests a phenomenological explanation for the observation that an agent with twice the compute of its opponent seems to win with a probability of roughly $2/3$, which in fact corresponds to a compute exponent of $\alpha_C = 1$. Similarly, Liu et al. (2021) report Elo scores that appear to scale as a log of environment frames for humanoid agents playing football, which would correspond to a power-law exponent of roughly 0.5 for playing strength scaling with data. Lee et al. (2022) apply the Transformer architecture to Atari games and plot performance scaling with the number of model parameters. Due to the substantially increased cost of calculating model-size scaling compared to compute or dataset-size scaling, they obtain only a limited number of data points, each generated by a single training seed. On this ba-

[1] https://github.com/OrenNeumann/AlphaZero-scaling-laws

sis, analyzing the scaling behavior seems difficult. First indications of the model-size scaling law presented in this work can be found in Neumann & Gros (2022).

## 3 BACKGROUND

### 3.1 LANGUAGE MODEL SCALING LAWS

Kaplan et al. (2020) showed that the cross-entropy loss of autoregressive Transformers (Vaswani et al., 2017) scales as a power law with model size, dataset size and compute. These scaling laws hold when training is not bottlenecked by the other two resources. Specifically, the model size power law applies when models are trained to convergence, while the compute scaling law is valid when training optimal-sized models. By combining these laws a power-law scaling of optimal model size with compute is obtained, with an exponent derived from the loss scaling exponents. These exponents, later recalculated by Hoffmann et al. (2022), tend to fall in the range $[0, 1]$.

### 3.2 POWER LAW SCALING AND ELO RATING

The Elo rating system (Elo, 1978) is a popular standard for benchmarking of MARL algorithms, as well as for rating human players in zero sum games. The ratings $r_i$ are calculated to fit the expected score of player $i$ in games against player $j$:

$$E_i = \frac{1}{1 + 10^{(r_j - r_i)/400}} \, , \tag{1}$$

where possible game scores are $\{0, 0.5, 1\}$ for a loss/draw/win respectively. This rating system is built on the assumption that game statistics adhere to the Bradley-Terry model (Bradley & Terry, 1952), for which each player $i$ is assigned a number $\gamma_i$ representing their strength. The player-specific strength determines the expected game outcomes according to:

$$E_i = \frac{\gamma_i}{\gamma_i + \gamma_j} \, . \tag{2}$$

Elo rating is a log-scale representation of player strengths (Coulom, 2007): $r_i = 400 \log_{10}(\gamma_i)$. An observed logarithmic scaling of the Elo score is hence equivalent to a power-law scaling of the individual strengths: If there exists a constant $c$ such that $r_i = c \cdot \log_{10}(X_i)$ for some variable $X_i$ (e.g. number of parameters), then the playing strength of player $i$ scales as a power of $X_i$:

$$\gamma_i \propto X_i^\alpha \, , \tag{3}$$

where $\alpha = c/400$. Note that multiplying $\gamma_i$ by a constant is equivalent to adding a constant to the Elo score $r_i$, which would not change the predicted game outcomes. This power law produces a simple expression for the expected result of a game between $i$ and $j$:

$$E_i = \frac{1}{1 + (X_j/X_i)^\alpha} \, . \tag{4}$$

## 4 EXPERIMENTAL SETTING

We train agents with the AlphaZero algorithm using Monte Carlo tree search (MCTS) (Coulom, 2006; Browne et al., 2012) guided by a multilayer perceptron. Kaplan et al. (2020) have observed

Table 1: Summary of scaling exponents. $\alpha_N$ describes the scaling of performance with model size, $\alpha_C$ of performance with compute, and $\alpha_C^{opt}$ of the optimal model size with compute. Connect Four and Pentago have nearly identical values.

| Exponents | Connect Four | Pentago |
|---|---|---|
| $\alpha_N$ | 0.88 | 0.87 |
| $\alpha_C$ | 0.55 | 0.55 |
| $\alpha_C^{opt}$ | 0.62 | 0.63 |

that autoregressive Transformer models display the same size scaling trend regardless of shape details, provided the number of layers is at least two. We therefore use a neural network architecture where the policy and value heads, each containing a fully-connected hidden layer, are mounted on a torso of two fully-connected hidden layers. All hidden layers are of equal width, which is varied between 4 and 256 neurons. Training is done using the AlphaZero Python implementation available in OpenSpiel (Lanctot et al., 2019).

In order to make our analysis as general as possible we specifically avoid hyperparameter tuning whenever possible, fixing most hyperparameters to the ones suggested in OpenSpiel's AlphaZero example code, see Appendix A. The only parameter tailored to each board game is the temperature drop, as we find that it has a substantial influence on agents' ability to learn effectively within a span of $10^4$ training steps, the number used in our simulations.

We focus on two different games, Connect Four and Pentago, which are both popular two-player zero-sum open-information games. In Connect Four, players drop tokens in turn into a vertical board, trying to win by connecting four of their tokens in a line. In Pentago, players place tokens on a board in an attempt to connect a line of five, with each turn ending with a rotation of one of the four quadrants of the board. These two games are non-trivial to learn and light enough to allow for training a larger number of agents with a reasonable amount of resources. Furthermore, they allow for the benchmarking of the trained agents against solvers that are either perfect (for Connect Four) or near-to-perfect (for Pentago). Our analysis starts with Connect Four, for which we train agents with varying model sizes. Elo scores are evaluated both within the group of trained agents and with respect to an open source game solver (Pons, 2015). We follow up the Connect Four results with agents trained on Pentago, which has a much larger branching factor and shorter games (on average, see Table 2).

For both games we train AlphaZero agents with different neural network sizes and/or distinct compute budgets, repeating each training six times with different random seeds to ensure reproducibility (Henderson et al., 2018). We run matches with all trained agents, in order to calculate the Elo ratings of the total pools of 1326 and 714 agents for Connect Four and Pentago, respectively. Elo score calculation is done using BayesElo (Coulom, 2008).

## 5 RESULTS

### 5.1 NEURAL NETWORK SIZE SCALING

Fig. 2 (top) shows the improvement of Connect Four Elo scores when increasing neural network sizes across several orders of magnitude. We look at the infinite compute limit, in the sense that performance is not bottlenecked by compute-time limitations, by training all agents exactly $10^4$ optimization steps. One observes that Elo scores follow a clear logarithmic trend which breaks only when approaching the hard boundary of perfect play. In order to verify that the observed plateau in Elo is indeed the maximal rating achievable, we plot the Elo difference between each agent and an optimal player guided by a game solver (Pons, 2015) using alpha-beta pruning (Knuth & Moore, 1975). A vertical line in both plots marks the point beyond which we exclude data from the fit. This is also the point where agents get within 10 Elo points difference from the solver.

Combining the logarithmic fit with Eq. 1 yields a simple relation between the expected game score $E_i$ for player $i$ against player $j$ and the ratio $N_i/N_j$, of the respective numbers of neural network

Table 2: Game details. Connect Four games last longer on average, but the branching factor of Pentago is substantially larger. Game lengths are averaged over all training runs, citing the maximal allowed number of turns as well.

| Game | Branching Factor | Average Game Length | Maximal Game Length |
|---|---|---|---|
| Connect Four | $\leq 7$ | 25.4 | 42 |
| Pentago | $\leq 288$ | 16.8 | 36 |

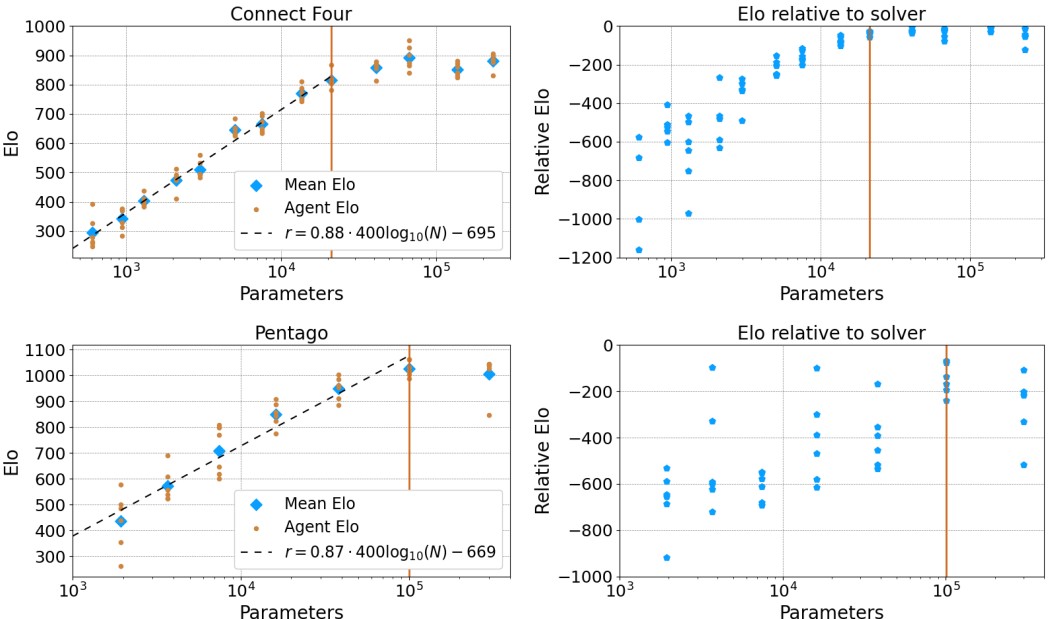

Figure 2: AlphaZero performance scaling with neural-network parameters. **Top left:** Agents trained on Connect Four. Elo scores follow a clear linear trend in log scale that breaks off only when approaching the optimal playing strategy. This can can be seen using a game solver (top right). **Top right:** Elo scores obtained by only considering matches against an optimal player, created with a game solver (Pons, 2015). Agents to the right of the vertical line are within ten Elo-points distance from the perfect solver. Increased data scattering is due to Elo scores being calculated against a single benchmark player. **Bottom left:** The Elo scores of Pentago as a function of model size follow the same trend as Connect Four. Improvement flattens for large parameter counts when performance approaches the level of a benchmark agent aided by an optimal play database (Irving, 2014) (bottom right).

parameters:

$$E_i = \frac{1}{1 + (N_j/N_i)^{\alpha_N}} , \tag{5}$$

where the power $\alpha_N$ is a constant determined by the slope of the fit. In the context of the Bradley-Terry model, Eq. (2), this means that player strength follows a power law in model size, $\gamma \propto N^{\alpha_N}$.

Results for the game of Pentago are presented at the bottom of Fig. 2. Pentago similarly exhibits log scaling of average agent Elo scores as a function of model size that flattens out only with a high number of neural network parameters. The scaling exponent is strikingly close to that of Connect Four (see Table 1), suggesting this exponent might be characteristic for a certain, yet unknown class of games. Again, in order to verify that the observed plateau is due to approaching the boundary of optimal play, we plot the Elo difference between each agent and a close to optimal benchmark player aided by a game solver. The bottom right of Fig. 2 shows that large agents are much closer to the level of the benchmark player. The benchmark player is based on the best performing Pentago agent trained, given 10 times more MCTS steps per move. We further improve it by constraining all moves before ply 18 to be optimal, using a 4TB dataset created by a Pentago solver program (Irving, 2014). A vertical line in both plots marks the point the log trend breaks, where agents approach the level of the benchmark player. Similar to Connect four we exclude the largest agent size beyond that point from the fit where adding more parameters no longer increases performance.

The power-law scaling of the playing strength of MARL agents can be considered analogous to the scaling laws observed for supervised learning. A noteworthy difference is the type of performance parameter being scaled. Scaling laws for language models tend to scale cross-entropy test loss with resources, which has been correlated to improved performance on downstream tasks (Hoffmann et al., 2022). In comparison, Eq. 5 shows power law scaling of player strength rather than test loss,

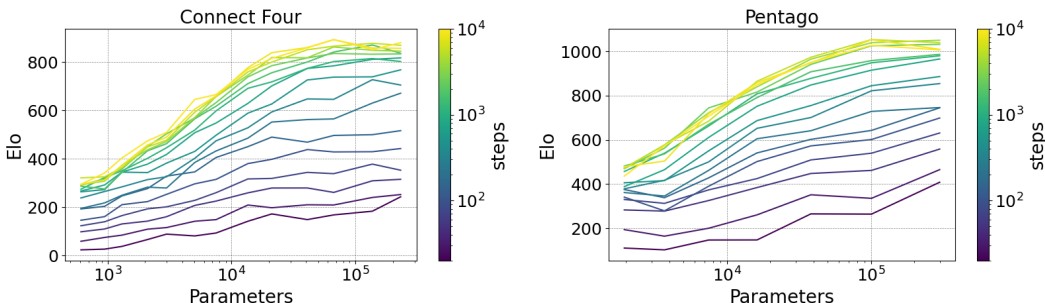

Figure 3: Elo scaling with parameters during training. For both games, log-linear scaling appears already at early stages of training, albeit with lower slopes. Elo curves converge for large numbers of training steps when agents approach their maximal potential.

which is directly related to the Elo score, a common measure of task performance in MARL. In fact, neural network loss on a test set generated by a game solver are observed to be a poor predictor of performance scaling both for the value head and the policy head, see Appendix F.

## 5.2 EFFECT OF TRAINING TIME ON SIZE SCALING

Similar to language model scaling laws, the scaling exponent of Eq. 5 only applies at the limit of infinite compute when training is not bottlenecked by the number of training steps. Fig. 3 shows how performance scaling is influenced by the amount of available compute in terms of training steps. One observes that the Elo score scales logarithmically with model size for all training regimes when performance is not affected, as before, by the perfect play barrier. The slope increases with the number of training steps, but with diminishing returns, see Appendix D. For both games one observes that Elo scores converge to a limiting curve when the compute budget becomes large. There is hence a limiting value for the scaling exponent $\alpha_N$, as defined by Eq. (5).

## 5.3 COMPUTE SCALING

To examine compute scaling we plot Elo scores against training self-play compute, defined here by $C = S \cdot T \cdot F \cdot D$, where $S$ is the number of optimization steps, $T$ is the number of MCTS simulations per move, $F$ the compute cost of a single neural network forward pass, and $D$ is the amount of new datapoints needed to trigger an optimization step. We do not count the compute spent during optimization steps, which is much smaller than the compute needed to generate training data. In our case we have approximately 2000 forward passes for each backward pass. We also approximate the number of MCTS simulations $T$ to be the maximum number of simulations allowed, since the maximum is reached at all game positions except late-game positions, where the remaining game tree is already fully mapped.

Fig. 4 shows the Elo scores of agents with different training compute measured in units of floating-point operations (FLOPs). The Pareto front, which is the group of all agents with maximum Elo among agents with similar or less compute, follows a log-linear trend up to the point of optimal play. Together with Eq. 1 this yields an expected game score of:

$$E_i = \frac{1}{1 + (C_j/C_i)^{\alpha_C}} \,, \tag{6}$$

for agents $i$, $j$ with training computes $C_i$, $C_j$ and optimal neural network sizes. Player strength scales with the same exponent, $\gamma \propto C^{\alpha_C}$. This scaling behaviour also holds for the average agent performance when we plot the Pareto front of compute curves averaged over multiple training runs with different random seeds (see Appendix D). Eq. 6 therefore applies not only to the best case training scenario, but also when training a single agent instance.

As with the size scaling exponent, Connect Four and Pentago have similar compute scaling exponents (see Table 1). We note that our training compute exponents are significantly smaller than the

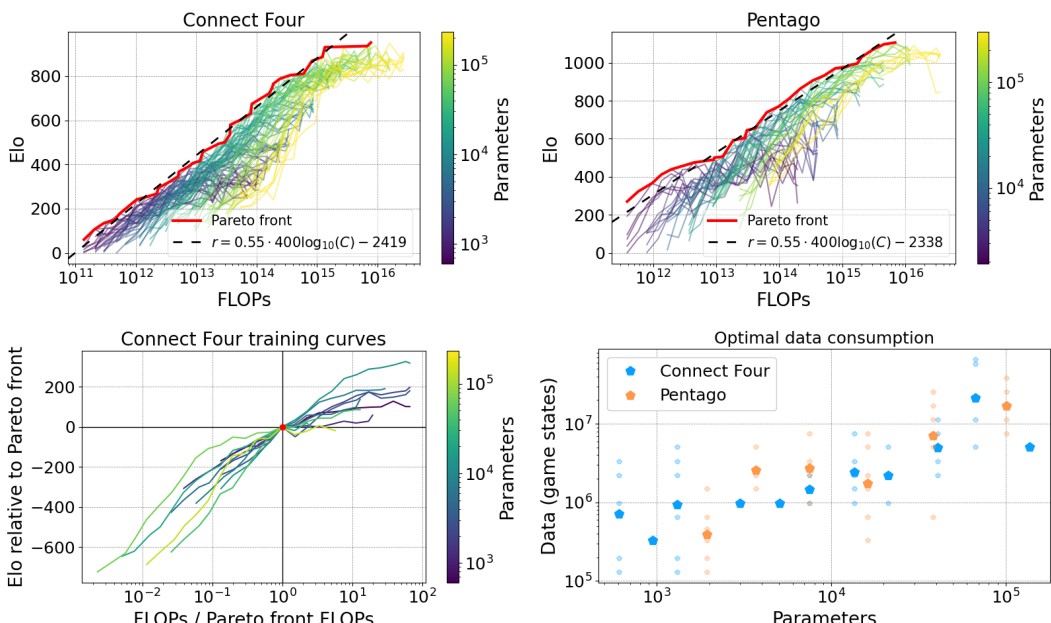

Figure 4: **Top:** Performance scaling with training compute. The Pareto front follows a log-linear trend in compute right up to the optimal play plateau, both for Connect Four and Pentago (left/right panel). **Bottom left:** Average Connect Four training curves relative to the point at which they reach the Pareto front. Agents continue to improve significantly after passing through the Pareto optimal point, meaning optimal training should stop long before convergence (compare to the training curves in Silver et al. (2017b;a)). **Bottom right:** The amount of data required to reach the compute Pareto front increases as the parameter count increases. Large dots are the geometric mean of data usage on the Pareto front.

compute exponent obtained when applying Eq. 6 to the data obtained by Jones (2021) for small-board Hex, which would be roughly $\alpha_C = 1.3$. The reason for this difference is at present unclear. Possible explanations could be the different game types, or the choice of neural net architecture; while we keep the depth constant, Jones (2021) increase both depth and width simultaneously.

## 5.4 ESTIMATING THE COMPUTE OPTIMAL MODEL SIZE

We now turn to our central result, efficiency scaling. Scaling laws are attracting considerable attention as they provide simple rules for the optimal balancing between network complexity and computing resources. The question is, to be precise, whether we can predict how many neural network parameters one should use when constrained by a certain compute budget in order to get an AlphaZero agent with a maximal Elo score.

The agents on the Pareto front shown in Fig. 4 have the highest Elo score for all agents with the same or lower compute budget. In order to find the relationship between optimal network size and the given compute budget, we examine the network size of the Pareto agents and the amount of FLOPs they consumed in training. This is shown in Fig. 1, which contains an aggregate of all compute-optimal agents trained on Connect Four and Pentago. We see a clear power law trend common to both games of the form:

$$N_{opt}(C) = \left(\frac{C}{C_0}\right)^{\alpha_C^{opt}},\qquad(7)$$

where $N_{opt}$ is the optimal neural network size and $\alpha_C^{opt}$, $C_0$ are constants.

Next we point out that the optimality scaling exponent $\alpha_C^{opt}$ can be calculated analytically given the scaling laws we demonstrated for parameter count and compute. For this we use Eq. (5) and Eq. (6).

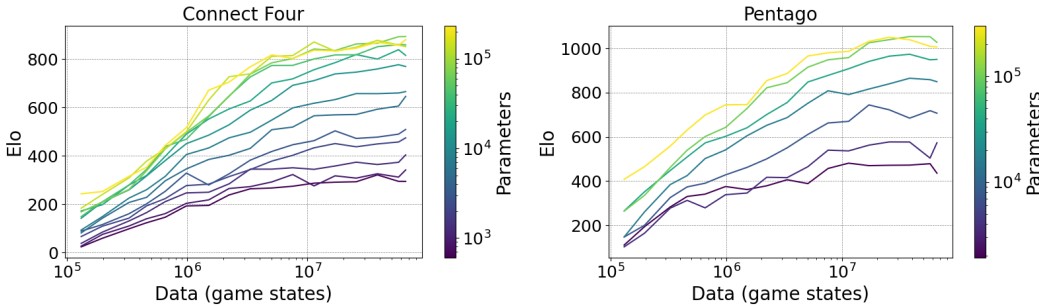

Figure 5: Performance improvement with data (game states) generated. Model sizes are color coded. Larger models are more sample efficient, achieving better Elo scores than smaller models when training on the same amount of game states.

Based on the fact that the Bradley Terry playing strength $\gamma$ scales as $\lim_{C \to \infty} \gamma(N, C) \propto N^{\alpha_N}$ at the limit of infinite compute, and as $\gamma(N_{opt}, C) \propto C^{\alpha_C}$ along the curve of optimal sizes, we can combine the two to obtain the constraint:

$$\alpha_C^{opt} = \frac{\alpha_C}{\alpha_N} \,. \tag{8}$$

This relationship is obtained when $N = N_{opt}$ and $C \to \infty$, but holds for every choice of $N$ or $C$ since the exponents are constants. The scale constant $C_0$ is calculated empirically. We use the scaling exponents $\alpha_N$ and $\alpha_C$ obtained by fitting the Elo ratings for Connect Four and Pentago to get $\alpha_C^{opt}$ via Eq. (8). The predicted scaling law is included in Fig. 1 fitting only $C_0$. The predicted exponent fits the data remarkably well.

### 5.4.1 EXTRAPOLATING TO STATE OF THE ART MODELS

For comparison, Fig. 1 right shows where AlphaGo Zero, trained by Silver et al. (2017b), and AlphaZero, trained by Silver et al. (2017a), would stand relative to the power law specified by Eq. (8). We base our estimates for the number of parameters from the respective papers, and use the training compute values estimated by Sevilla et al. (2022) and Amodei et al. (2018) for these models. We find that these models fall well below the optimal curve predicted from scaling MLP agents playing Connect Four and Pentago.

The scaling law plotted in Fig. 1 might not accurately describe AlphaGo Zero and AlphaZero trained on Go, chess and shogi due to the differences between the games: these games are much larger in observation size and branching factor than Connect Four and Pentago, as well as having different win conditions and game mechanics. The use of ResNets rather than MLPs can also be significant. It remains to future study to determine if these models exhibit related scaling trends. In the case that Eq. (8) does apply to agents using ResNets and trained on Go, chess, or shogi, then a better Elo score could have been achieved with the same training time by training larger neural nets by up to two orders of magnitude. A possible indication that these models were too small for the amount of compute spent training them can be seen from the training figures of Silver et al. (2017b;a); the AlphaGo Zero and AlphaZero training curves both have long tails with minimal improvement at late stages of training. Our analysis shows that optimal training for a fixed compute budget will stop long before performance has converged, in agreement with language model scaling laws (Kaplan et al., 2020), compare Fig. 4 bottom left. That being said, a reason to favour smaller models is their lower inference-time compute cost, which can be critical in the case of a time limit on move selection. We elaborate on inference compute scaling in Appendix B.

### 5.5 SAMPLE EFFICIENCY

Unlike supervised learning models where dataset size is an important metric due to the cost of collecting it, AlphaZero is not constrained by data limitations since it generates its own data while training. It is however important to understand how data requirements change as it provides us an insight into other sub-fields of reinforcement learning where data generation is costly, for example

in robotics models, which need real life interactions with the environment in order to gather training data (Dulac-Arnold et al., 2021).

For this reason we look at performance scaling with data for different sized models, as shown in Fig. 5. We observe that agents improve at different rates with the amount of training data, which is defined here by the number of game states seen during self-play. Specifically, agents with larger neural nets consistently outperform smaller agents trained on the same amount of data, a phenomenon of data efficiency that has been observed in language models (Kaplan et al., 2020). However, when training is not bottlenecked by the amount of training data, an opposite trend appears: the amount of data required to reach optimal performance under a fixed compute budget increases with model size, as is visible in the bottom right panel of Fig. 4.

## 6   DISCUSSION

In this work we provided evidence to the existence of power-law scaling for the performance of agents using the AlphaZero algorithm on two games, Connect Four and Pentago. For both games one observes scaling laws qualitatively paralleling the ones found in neural language models. We pointed out that a log-scaling of Elo scores implies a power law in playing strengths, a relation that is captured by Eq. (4) and that can be used to predict game results. We showed that AlphaZero agents that are trained to convergence, with no limit on available compute, achieve Elo scores that are proportional to the logarithm of the respective number of neural network parameters. We demonstrated this for two games with significantly different branching parameters and tyical game lengths, finding that they share similar scaling exponents for playing strength. This result indicates the possible existence of universality classes for performance scaling in two-player open information games.

We then demonstrated that Elo scores scale logarithmically with available compute when training optimally sized agents, again finding similar scaling exponents for the two games studied. We showed that the optimal number of neural network parameters scales as a power of compute with an exponent that can be derived from the other two scaling laws. This scaling model is especially important as it allows one to achieve maximal performance within the limits of a given training budget by using an optimally sized neural net. We showed optimal training stops long before convergence, in contrast to what was done with state-of-the-art models. Finally we observed that agents with larger neural nets are always more data efficient than smaller models, which could be important in particular for reinforcement learning studies with expensive data generation.

We find it note worthy that scaling laws that are common to language and other supervised learning models are also present in one of the most important MARL models. This scaling behavior could be common to other reinforcement learning algorithms, which would provide an opportunity to optimize their resource allocation. We note that the main significance of the scaling exponents found in this study is not their reported value, but the evidence of power-law scaling. Consider the scaling exponents found by Kaplan et al. (2020), which were later corrected by optimizing the learning rate schedule (Hoffmann et al., 2022). The tuning of AlphaZero's hyperparameters could affect the observed exponents in a similar manner. Particularly important hyperparameters are the number of states explored by the MCTS, the frequency of optimization steps, and the update frequency of the data generating model. We leave the investigation of hyperparameter impact to future work.

As a side remark, we would like to comment on the relevance of scaling laws for studies without massive computing budgets. A rising concern since the discovery of performance scaling laws regards the resulting push towards substantially increased model sizes. This decreases the impact of research utilizing smaller-sized models, the bread-and-butter approach of smaller research groups. (Strubell et al., 2019). On the other hand, the existence of scaling laws implies that it is possible to gain relevant insights into the behaviour of large models by training medium-sized models, the case of the present work. Interestingly, our initial test runs were performed on a toy problem that involved training agents on a desktop computer with just four CPU cores. Even with such a minimal compute budget a clear scaling trend emerged (Neumann & Gros, 2021). We believe our work shows that significant insights can be obtained at times also with small-scale analysis.

ACKNOWLEDGMENTS

We would like to thank Ivo FD Oliveira and Geoffrey Irving for helpful discussions and feedback.

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

## A  Hyperparameters

In order to keep our results as general as possible, we try to avoid hyperparameter tuning and choose to train all agents with the values suggested by OpenSpiel's Python AlphaZero example. Table 3 contains a summary of all hyperparameters used and their meaning.

Table 3: Training hyperparameters used for Connect Four, Pentago and Oware.

| Hyperparameter | Value | Description |
|---|---|---|
| $c_{uct}$ | 2 | MCTS exploration constant |
| Max simulations | 300 | Number of MCTS steps to run per move |
| Batch size | $2^{10}$ | Batch size during optimization steps |
| Replay-buffer size | $2^{16}$ | Number of game states stored in the replay buffer |
| Replay-buffer reuse | 10 | Number of optimization steps a game state stays in the buffer before it is erased |
| Learning rate | 0.001 | Optimization learning rate |
| Weight decay | 0.0001 | L2 regularization strength |
| Policy $\epsilon$ | 0.25 | Policy noise |
| Policy $\alpha$ | 1 | Dirichlet noise variable |
| Temperature | 1 | Temperature applied to the policy for final move selection |
| Temperature drop | varied | Number of turns before temperature is set to 0 |

*Temperature drop* is the only parameter with a different value in each game, set to 15 and 5 for Connect Four and Pentago, respectively. We do this since we find that this term, which should correlate with game length, can drastically affect the performance of agents when we train them for $10^4$ optimization steps.

*Replay-buffer reuse* is set to 10 rather than 3, the value suggested in the original example, to speed up training by leaving states for longer in the replay buffer (Lillicrap et al., 2015).

During matches, all noise is set to zero but temperature is set to $0.25$, with an infinite *temperature drop*, meaning temperature stays constant throughout the game. We do this since without a random element all games between two agents will be identical. A high temperature would distort the data

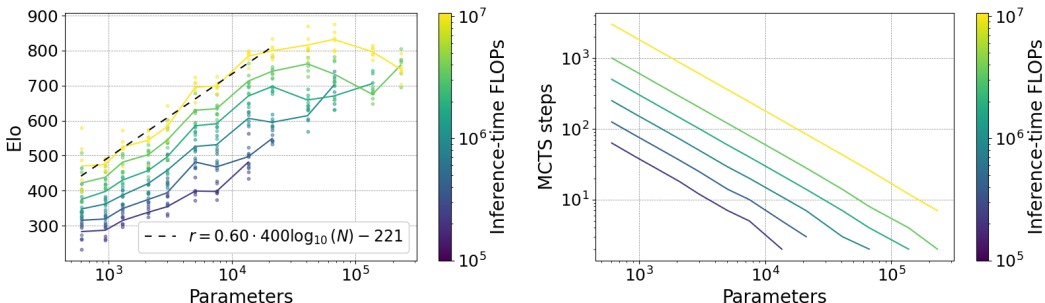

Figure 6: **Left:** Connect Four performance at fixed values of inference-time compute, enforced by scaling MCTS search size inversely with forward-pass compute costs. Elo scores steadily improve with increasing model size for smaller models, but the trend breaks for large models. **Right:** The number of MCTS steps used by different sized models with different inference-time compute budgets. Agents with less than 2 MCTS steps per turn are excluded from both plots.

by reducing the Elo difference between players, which is why we use a low temperature value of $0.25$ that produces roughly $5\%$ or less repeating games. The number of MCTS simulations per move is the same for all agents at test time, equal to the number used during training.

## B    INFERENCE-TIME COMPUTE SCALING

Reinforcement learning models like AlphaZero are often required to perform fast at test time, either in order to finish their calculation within a thinking time limit or to react quickly to a rapidly changing environment. The AlphaZero algorithm provides the ability to tailor the move-selection inference time to a given time limit by changing the number of MCTS steps, which does not have to match the number of steps used during training. Since our main results all use a fixed number of 300 MCTS steps both at train and test time, we add here the scaling behavior of different sized Connect Four agents using varying numbers of MCTS steps. We look at the case where training compute is abundant but performance is bottlenecked by inference-time compute. To do so we match fully converged agents against each other, all receiving a fixed amount of inference-time compute enforced by scaling MCTS search size inversely with forward-pass compute costs.

Fig. 6 shows the Elo scaling of different sized agents under inference-time compute constraints. Agents steadily improve with size at smaller sizes for all amounts of available inference-time compute. Larger agents break off from this trend at different points which depend on the amount of compute given, such that the the trend seems to hold longer for larger compute values. Interestingly, a power law appears at the limit of large inference-time compute which breaks off only for the largest agents. The scaling exponent is significantly smaller than that of Fig. 2, which is not surprising given that small agents have access to up to two orders of magnitude more MCTS steps than the largest agents, see Fig. 6 right. This scaling law may be related to the power law scaling of inference-time compute with training compute found by Jones (2021) when fixing the Elo score of AlphaZero agents.

## C    MEASURING ELO AGAINST OTHER PLAYERS

Elo rating is a system that generates predictions of game outcomes that are based on ground truth values. As such, it is vulnerable to biases in the underlying dataset of game outcomes. One example of a potential bias is when all players use a restricted set of strategies, resulting in Elo scores that will not necessarily represent the interactions of these players with agents utilizing different sets of strategies. Here we provide evidence that the scaling laws presented in this paper are not the result of such biases.

Although Fig. 2 right presents Elo scores calculated solely from matches between agents and a perfect Connect Four game solver, these scores are only accurate close to the solver score and are unreliable for agents far from optimal play. For example, the worst agent is almost $1{,}200$ Elo points

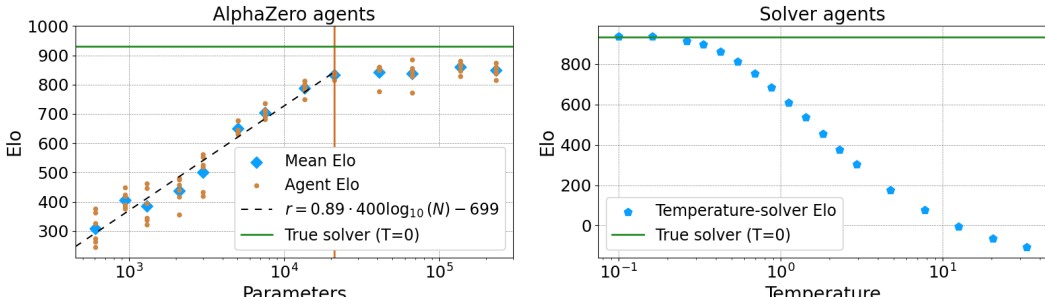

Figure 7: **Left:** AlphaZero size scaling with Elo scores calculated by only using matches between solver agents and AlphaZero agents, or matches between two solvers. The scaling behavior of Fig. 2 holds here as well, with an almost identical scaling exponent. Since the true solver, with temperature set to zero, is part of the agent pool, it has a well defined Elo score (green line). **Right:** Solver-agent scaling with temperature. Each solver agent uses the prior produced by the Connect Four game solver (Pons, 2015), applying a temperature transformation to it according to Eq. 9. Low temperature agents asymptotically approach the true solver Elo score (green line), while high temperature agents converge to the score of a random player. The range of Elo scores covers that of AlphaZero agents, allowing an accurate measurement of their Elo scores.

away from the solver, meaning it should win about one game in a 1,000. This requires a large amount of matches in order to model accurately, and indeed several agents are missing from this figure since they received an Elo score of minus infinity, never winning a single match.

Here we solve this issue by producing a set of solver agents of varying playing strengths, matching them with each other and with AlphaZero agents trained on Connect Four to produce Fig. 7. All solver agents base their prior on the vector $q$ provided by the game solver, which contains information about the quality of all legal moves. Taking action $a$ in game state $s$, the number $q(s, a)$ equals to 42 minus the minimal number of turns until either player can force a victory under perfect play, with a minus sign if the current player would lose. See Pons (2015) for a detailed explanation of how $q$ is generated. The final prior $\pi$ is created by applying a softmax to $q$ divided by a temperature value $T$:

$$\pi(s, a) = \frac{e^{q(s,a)/T}}{\sum_b e^{q(s,b)/T}} .$$
(9)

This provides a smooth interpolation between the optimal prior and a uniform prior. At $T = 0$ the prior leads to the fastest victory if possible, otherwise to a draw if possible, and if neither then to the slowest defeat. At the limit $T \rightarrow \infty$ the prior is uniform, generating a completely random agent. Note that $T$ is *not* the temperature mentioned in Table 3, which is set to zero for the solver agents and to 0.25 for AlphaZero agents, as it is done throughout this paper, see Appendix A.

Fig. 7 shows the Elo scaling calculated with solver agents. Unlike in Fig. 2, Elo scores are calculated exclusively from data on matches between AlphaZero agents and solver agents as well as matches between pairs of solver agents. No game between two AlphaZero agents was included in this calculation. The resulting Elo scores scale as a log of model size with an almost identical exponent to that found in Table 1, providing evidence that this scaling behavior is universal and not the product of a bias in the choice of players.

We note that while this way of measuring Elo strengthens our results by reproducing the scaling exponents, it is probably a less accurate measure than the one used in Section 5.1. This is because all solver agents use the same strategy, which introduces bias into the match outcome dataset. This could be the reason why the spread of agent Elo scores in Fig. 7 is slightly wider than that of Fig. 2 for smaller agents. Despite this bias, the model size scaling law seems robust enough to appear here as well.

## D SUPPLEMENTAL FIGURES

### D.1 AVERAGE COMPUTE SCALING

The scaling law presented in Fig. 4 is obtained by finding the Pareto optimal agents among all agents trained. It does not make clear whether the same scaling law would apply when training a single agent, rather than training with several random seeds and picking the best one. We therefore plot Fig. 8, which displays Elo scaling with compute, but with scores averaged over each neural-net size group. The scaling exponents deviate only slightly from the exponents calculated with individual agent data. This implies one can expect to find compute log-scaling of Elo also when training single agents, as opposed to training many seeds and picking the best performing one.

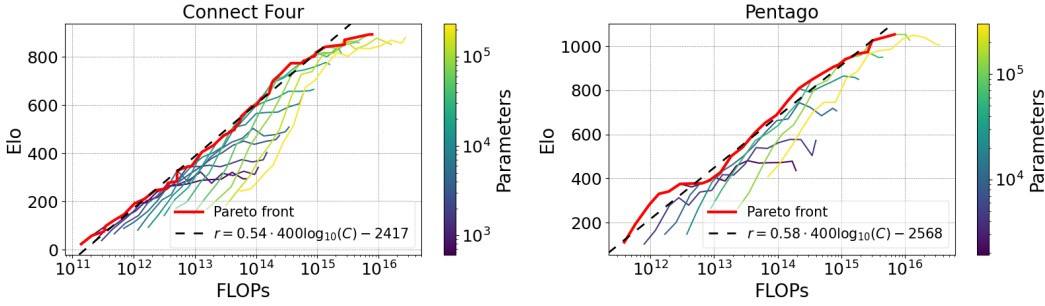

Figure 8: Averaged performance scaling with training compute. Re-plotting figure 4 with data averaged over 6 different seeds for each number of neural-net parameters produces a Pareto front with a similar scaling exponent.

### D.2 SIZE SCALING CONVERGENCE

The parameter-count scaling law presented in section 5.1 only holds in the limit of infinite compute, when agents are allowed to train to convergence. A way of measuring the proximity to this limit is by looking at the evolution of the log-linear fit to the data. We have shown in Fig 3 that log-linear scaling appears already at short training lengths, but with lower slopes. In Fig 9 we plot the exponent values derived from these slopes. One can see that the exponent value slowly converges to its final value at $10^4$ training steps.

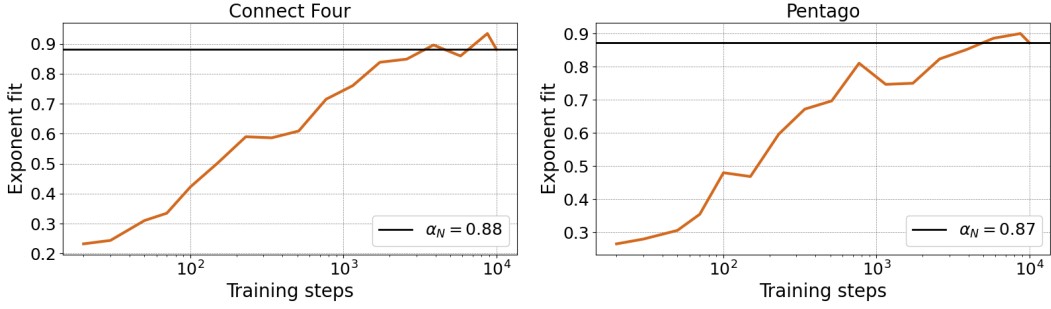

Figure 9: Convergence of the scaling exponent $\alpha_N$. Fitting the curves in Fig 3 we obtain intermediate values for the scaling exponent during training. The exponent fits grow larger with training length, slowing down towards a final value for both games.

## E OWARE SCALING

We measure the scaling laws of another popular board game, Oware. We choose this game because of its length; unlike Connect Four and Pentago, Oware games are not bound to end after a fixed

number of turns. This also means the average number of turns per game is much larger compared to the other games, see Table 4. In practice, infinite games are handled in OpenSpiel by calling them off after $1,000$ steps have been played. These games, although rare, are excluded from our statistics. Another difference between Oware and the other games is the shape of the observation tensor. For Oware, neural network input is in the shape of a 14-elements vector of integers, counting the number of seeds in each pit. This is unlike the observation tensor of Connect Four and Pentago as well as games like Go, where board state is presented in binary encoding.

We find that Oware exhibits power-law scaling in playing strength in neural network size and compute, but encounter two serious issues during training. One issue is the lack of sufficient training time for agents to converge, not allowing us to gauge correctly the true value of the size scaling exponent. The other issue is that the largest models do not manage to reach their full potential, getting Elo scores significantly lower than smaller models. We give a possible explanation to this in the last section. This issue mostly affects the correct estimation of the compute exponent.

Table 4: Game details including Oware. Oware games are not bounded to a maximal number of moves. As a result, they last much longer than Connect four or Pentago games on average. Additionaly, the neural network input comes in the form of integer numbers in Oware, rather than zeros and ones.

| Game | Branching Factor | Average Game Length | Observation Shape |
| --- | --- | --- | --- |
| Connect Four | $\leq 7$ | 25.4 (max. 42) | $3 \times 6 \times 7$ Boolean tensor |
| Pentago | $\leq 288$ | 16.8 (max. 36) | $3 \times 6 \times 6$ Boolean tensor |
| Oware | $\leq 6$ | 100 (unbounded) | 14-long integer vector |

### E.1 SIZE SCALING

We repeat the analysis of neural network parameter scaling for Oware agents. Fig. 10 presents the scaling of Elo scores with model size. For Oware one observes, interestingly, that agents with large numbers of parameters do not train to optimality. Small- and medium-sized agents show however evidence of power-law scaling. It is clear that large agents could in theory reach better performance, since the range of strategies a MLP network could implement is always contained within that of a larger model. We speculate about the cause to this in section E.4.

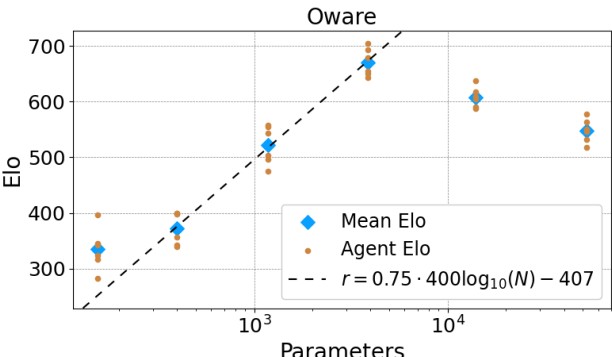

Figure 10: Oware size scaling. Oware shows a logarithmic scaling trend of Elo with neural net size that breaks abruptly for larger networks. Since the strategy space available to smaller agents is contained within that available to larger ones, large agents should at least reach an equal Elo score to smaller agents. It is therefore clear that the large agents did not converge to optimal performance.

Medium-sized agents performance follows a log scaling with model size, with a smaller scaling exponent than that found with Connect Four and Pentago. However when we plot the changing of the exponent fit during training in Fig. 11, it seems very likely that it will keep increasing with

longer training. Unfortunately we could not extend training length by another order of magnitude in the time given for this analysis.

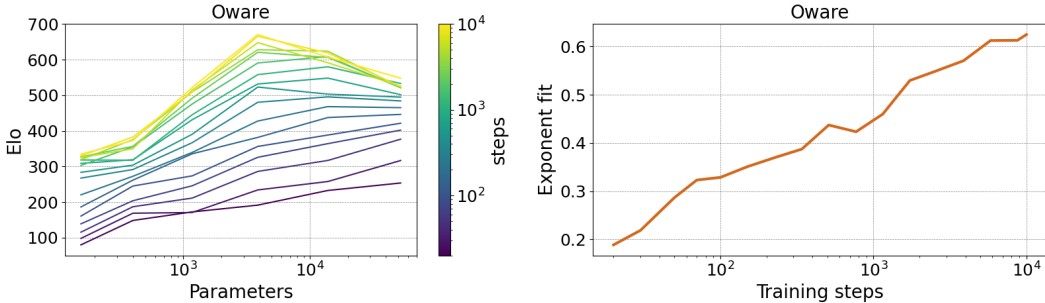

Figure 11: **Left:** Elo scaling with parameters during training. Log-linear scaling already appears at early stages of training, converging to a fixed curve. **Right:** $\alpha_N$ convergence. The exponent fit keeps increasing with training time for Oware, suggesting the final exponent should be larger.

### E.2  COMPUTE SCALING

Fig. 12 shows the Elo scores of Oware agents with different training compute. Despite the failure of large agents to learn effectively, Oware matches a log-linear fit over several orders of magnitude with a smaller exponent than that found in Connect Four and Pentago.

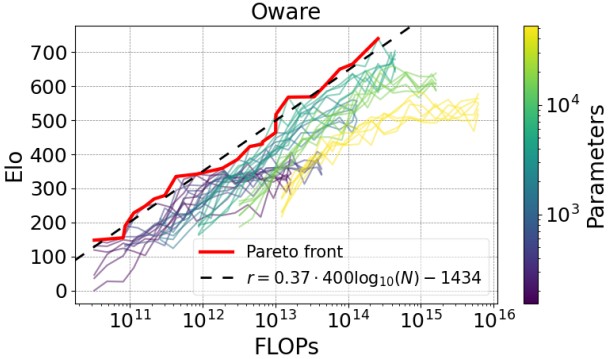

Figure 12: Oware performance scaling with training compute. The Pareto front follows a log-linear trend in compute. Large models, which failed to train well, do not make it to the Pareto front.

### E.3  ESTIMATING THE COMPUTE OPTIMAL MODEL SIZE

We use the exponents $\alpha_N$ and $\alpha_C$ found for Oware to fit a new scaling law for the optimal model size, $N_{opt}$. Since Oware has a smaller compute-scaling exponent than the other games, its optimality exponent $\alpha_C^{opt}$ is also smaller. Overlaying the resulting power law on data from other games in Fig. 13, we find that it does not fit Connect Four and Pentago well. Even though the Oware exponent is significantly smaller, it still predicts that AlphaGo Zero and AlphaZero would benefit from an increase in model size.

### E.4  TRAINING ISSUES

Reinforcement learning algorithms can exhibit convergence issues, as is evident in the case of Oware. The largest agents trained on this game clearly failed to converge to their optimal playing ability, since they under-performed smaller agents, which they should at least match in performance. We believe a likely cause to that is the format of the input tensor, which is composed of a vector of

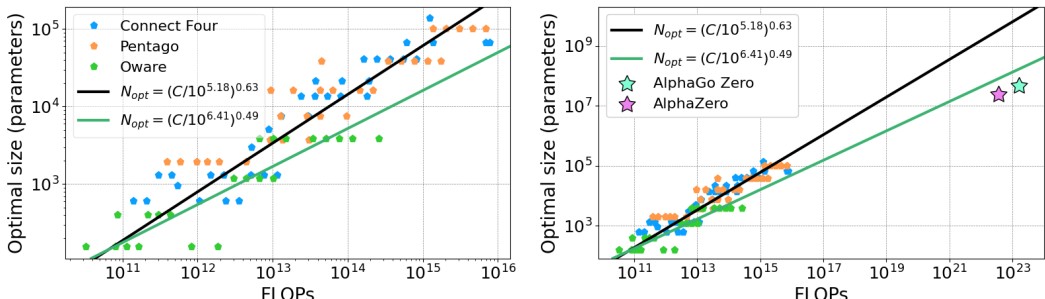

Figure 13: Optimal number of neural network parameters as in Fig. 1, with the scaling trend predicted with Oware in green. Even this lower exponent scaling law predicts previous state-of-the-art models were undersized, although to a much lesser degree.

integers rather than a boolean tensor as used in Connect Four and Pentago and games like Go and chess (see Table 4). We observe the same trend of large models consistently failing to train well in another game, a toy model problem not presented here with floating-point inputs rather than binary ones. We find that large models trained on the toy problem with no issues when we use a modified version of AlphaZero created by us, which imports a key feature from AlphaGo Zero. Specifically, the modified version tests each new candidate for the data generating model against the current model, and only updates it if the new candidate is proved to perform better. OpenSpiel currently does not contain this feature, but we believe it could be useful in the case of non-boolean input.

Table 5: Scaling exponent for all three games. Oware shows smaller size and compute exponents, and a larger optimal size scaling exponent.

| Exponents | Connect Four | Pentago | Oware |
|---|---|---|---|
| $\alpha_N$ | 0.88 | 0.87 | 0.75 |
| $\alpha_C$ | 0.55 | 0.55 | 0.37 |
| $\alpha_C^{opt}$ | 0.62 | 0.63 | 0.49 |

## F    TEST SET LOSS

Most language model scaling laws revolve around scaling test-set loss as a power law of some quantity. In contrast, scaling laws for AlphaZero appear in the form of playing strength power-laws. When we look at the predictive ability of trained agents on a test dataset, we find that test loss does not scale as a power law.

To measure test loss scaling, we create a dataset of game states from random Connect Four games generated by taking random moves. We then annotate these states with a perfect solver (Pons, 2015). The solver provides us with the ground-truth state value ($1, 0, -1$ for win, draw loss from the current position, respectively) and a policy prior which is a uniform probability distribution over all optimal moves. Optimal moves are defined as those which lead to either the fastest victory, a draw otherwise, or the slowest loss otherwise. All sub-optimal moves are given probability zero.

We then compare the value estimations and policy priors created by the solver to those predicted by fully trained Connect Four agents, by calculating the value and policy losses. These are defined in the full AlphaZero loss function (Silver et al., 2017b):

$$l = (z - v)^2 - \boldsymbol{\pi}^\top \log \boldsymbol{p} + c||\theta||^2 \,, \tag{10}$$

where $z$ is the recorded game outcome from the current position and $\boldsymbol{\pi}$ is the improved policy generated by the MCTS using the visit count of each node. We replace $z$ and $\boldsymbol{\pi}$ with predictions made by the solver.

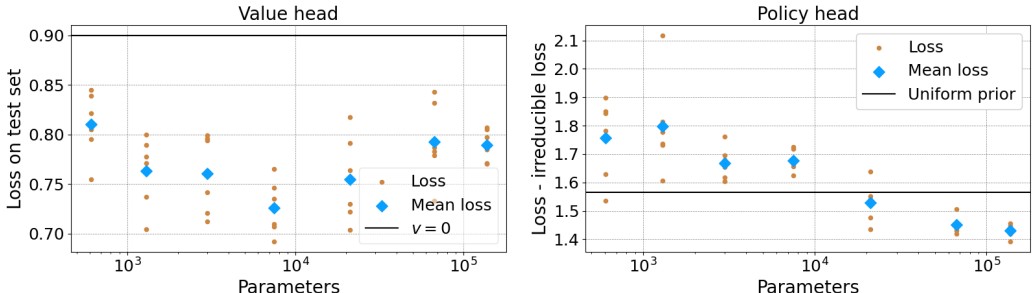

Figure 14: Test loss of Connect Four agents on a dataset created by a perfect solver. **Left:** Value head test loss. Loss changes very little with model size, large models have similar loss to small models. For comparison, we add a baseline loss received when always predicting a draw ($v = 0$). **Right:** Policy head test loss minus irreducible loss. Loss does decrease with size, indicating larger models produce a policy prior that is closer to the perfect policy. However the distance to the perfect prior is still large across all scales. For comparison, we add a baseline achieved when always predicting a uniform prior over all legal moves. Smaller agents achieve worse loss than the uniform baseline.

The resulting scaling trends appear in Fig. 14. For the policy head, we plot the test loss minus the irreducible loss (equal 0.336), the minimum loss achieved by matching the solver's policy prior exactly.

While policy loss does seem to decrease with model size, it is contained to a small range far from the minimal possible loss. Value loss decreases and then increases with model size. Both quantities do not seem to be good indicators of performance scaling.

We note that even though Elo score turns out to be a better measure of performance than test loss, there may exist even better measures. Elo rating fails to correctly model player dynamics when they are non-transitive, for example (Omidshafiei et al., 2019). Some alternatives exist (Omidshafiei et al., 2019; Oliveira et al., 2018), and it is possible they could capture elements of the agent dynamics that Elo rating would miss.

