# OpenReview forum: "Scaling Laws for a Multi-Agent Reinforcement Learning Model"
_ICLR.cc/2023/Conference — ICLR 2023 poster_

### Official Review · Reviewer_T3hx · 2022-10-21

**Confidence:** 4
**Correctness:** 1
**Technical Novelty And Significance:** 3
**Empirical Novelty And Significance:** 3
**Recommendation:** 6

**Clarity, Quality, Novelty And Reproducibility:**

The paper is well-written, clear and easy to follow.
The manuscript has high quality and I have relatively few concerns.
There is no novelty in the paper in terms of algorithmic changes, but overall the thorough analysis of the power-law scaling for MARL using ELO as the y-axis is novel.
The authors open-sourced the code for reproducing the experiments. Hence, there are no concerns regarding the reproducibly of results.

**Strength And Weaknesses:**

## Strengths

- The paper takes one of the most important issues of deep learning: how the changes in model size, compute, and data affect the performance of the model. This question has been studied extensively for NLP methods. However, not a lot of analysis exists for RL, particularly multi-agent domains. For me, the problem itself is real and practical.
- The idea of using ELO as the y-axis for RL scaling laws is interesting.
- This paper provides a comprehensive set of experiments, including ELO performances relative to the population of agents as well as solvers.

## Weaknesses

- While I appreciate that the authors have restrictions for running large-scale experiments, I do still think that the choice of games for this work is limited.
    - The authors use their results from Connect Four and Pentago to extrapolate insights about AlphaZero (Silver et al, 2017a) and AlphaGoZero (Silver et al, 2017b). This is an Apples to Oranges comparison involving different games and model architectures. I’d expect at least using the game dynamics (perhaps with smaller board sizes) before being able to comfortably generalise claims based on limited data.
    - For example, 8x8 checkers or smaller-sized Go would have strengthened the paper.
    - This would also allow authors to analyse bigger models that do not saturate as quickly.
    - With current results, it is unclear if such phenomena will be observed in more challenging games, such as Go and Chess.
- The ELO ranking results depend heavily on the population of agents being used for comparison. The authors use the ELO between the population of trained agents to estimate the power-law scaling in MARL. It is unclear if these insights can hold more generally. For example, the Relative ELO relative solver in Pentago does not seem to be as neat as the ELO relative to the agent population.
- The results suggest that the two games share exactly similar scaling exponents for playing strengths ($\alpha_N$, $\alpha_C$, and $\alpha_C^{opt}$). I find this quite surprising and unintuitive. This is also among the reasons why I’d like to see more games investigated in this game to assess the universality of these constants.
- The authors factor the compute by the forward pass costs and MCTC simulations. This is sensible, considering the model architecture used. Considering that the games are relatively easy, I’d be curious to see if such a power law emerges with model-free RL approaches.

**Summary Of The Paper:**

Inspired by the recent interest in power-law scaling of performances in large transformer-based models, this work investigates that in two-player zero-sum games with reinforcement learning (RL). By conducting experiments in two domains, Connect Four and Pentago games, authors show that the playing strengths of agents scale as a power law with the neural network size when models are trained until convergence. They also investigate the trade-off between model size and compute. Lastly, they utilise the two scaling laws to find a scaling law for the optimal model size given the amount of compute available.

**Summary Of The Review:**

Overall, this is an interesting paper that tackles an important problem, the scaling laws in a subclass of multi-agent RL methods. The evaluation protocol is novel and the paper includes a comprehensive set of experiments to support its main claims. That said, I still have some concerns about the generality of the results achieved here. The paper could have been improved by adding more games in the experimental setup. Furthermore, there are some concerns regarding the ELO-ranking population and scaling exponent values.

---

> ### Author Response · Authors · 2022-11-15
> **Reply to reviewer T3hx (1/2)**
>
>
> Thank you for the positive feedback. We are glad that you find our paper interesting, and see the problem we tackle as important and practical.
>
> We address the concerns raised one by one:
>
> > While I appreciate that the authors have restrictions for running large-scale experiments, I do still think that the choice of games for this work is limited.
>
> We agree that adding more games would have strengthened the paper. The exploration of scaling laws covering several orders of magnitude is however exceedingly demanding. Correctly calculating the size scaling laws in our paper demanded training several large networks to convergence, which is why we only used three relatively small board games. This is probably the reason why Kaplan et al. (2020) and Hoffmann et al. (2022) based their works on Transformers trained on a single dataset (with Hoffmann et al. adding a comparison of models trained on two subsets of the dataset in the appendix). Jones (2021) does the same, focusing on a single game in order to calculate AlphaZero compute scaling.
>
> > The authors use their results from Connect Four and Pentago to extrapolate insights about AlphaZero (Silver et al, 2017a) and AlphaGoZero (Silver et al, 2017b). This is an Apples to Oranges comparison involving different games and model architectures.
>
> We have changed all claims regarding extrapolation of our scaling laws to AlphaZero and AlphaGoZero, and made them more conservative. We also added a new section (5.4.1) where we explicitly detail the reasons why the scaling laws we found might not extrapolate to previous SOTA models. If this issue was the reason for the low correctness score given, we agree that our original formulation was indeed misleading. We believe that our main claim, that AlphaZero can exhibit power law scaling behaviors, is supported by solid evidence. We hope that the manuscript can be regarded now as correct.
>
> > The ELO ranking results depend heavily on the population of agents being used for comparison. The authors use the ELO between the population of trained agents to estimate the power-law scaling in MARL. It is unclear if these insights can hold more generally. For example, the Relative ELO relative solver in Pentago does not seem to be as neat as the ELO relative to the agent population.
>
> This is a valid concern, since Elo scores can definitely be affected by the type of players examined if they all use a limited set of strategies.
> To answer your concern, we ran new experiments using a new class of Connect Four agents based on a solver modified with temperature noise. We added the new experiments in the new appendix section C. By only using matches between AlphaZero agents and solver agents (and matches between solvers), we recalculate the Elo scores. We find that our scaling law remains valid and that the corresponding exponents are essentially identical.
>
> Regarding the point that the Elo-relative-to-solver plots of figure 2 don't look as nice as the main plots, we can provide a simple explanation for it. Elo rating is robust when it uses matches between many players, but these plots are based on matches between AlphaZero agents and a single solver agent. Other than the obvious bias of playing only against one player with one strategy, this means agents far from the solver have a very poor estimation of their Elo scores. This is because they require exponentially more matches in order to capture their smaller and smaller winning probabilities. We only added these plots so that one can compare which agents are close to the solver and which are far away.
>
> > The results suggest that the two games share exactly similar scaling exponents for playing strengths. I find this quite surprising and unintuitive. This is also among the reasons why I’d like to see more games investigated in this game to assess the universality of these constants.
>
> while we agree this result is surprising, we don't see it as unintuitive. Previous works on language and vision models have found scaling exponents which generalize across a range of distinct modalities. For example, Henighan et al. (2020) found the same optimal scaling exponent for fitting a language dataset, image dataset, video dataset, multimodal image to text and text to image as well as mathematical problem solving (see their figure 2). This universality of scaling exponents is one of the reasons they have attracted considerable attention recently. The existence of different RL environments that share the same scaling exponents is a novel discovery, but also one that matches results in parallel fields.

---

> > ### Author Response · Authors · 2022-11-15
> > **Reply to reviewer T3hx (2/2)**
> >
> >
> > > The authors factor the compute by the forward pass costs and MCTC simulations. This is sensible, considering the model architecture used. Considering that the games are relatively easy, I’d be curious to see if such a power law emerges with model-free RL approaches.
> >
> > We would also be curious to see how model-free agents like PPO and DQN scale. We intend to explore these cases in future works.
> >
> >
> > We hope we have addressed all your concerns, and would be glad to answer any more you may have.
> >
> >
> > ### References:
> >
> > * Kaplan, Jared, et al. "Scaling laws for neural language models." arXiv preprint arXiv:2001.08361 (2020).
> >
> > * Hoffmann, Jordan, et al. "Training Compute-Optimal Large Language Models." arXiv preprint arXiv:2203.15556 (2022).
> >
> > * Jones, Andy L. "Scaling scaling laws with board games." arXiv preprint arXiv:2104.03113 (2021).
> >
> > * Henighan, Tom, et al. "Scaling laws for autoregressive generative modeling." arXiv preprint arXiv:2010.14701 (2020).

---

> > ### Comment · Reviewer_T3hx · 2022-12-07
> > **Response to rebuttal**
> >
> > I thank the authors for their response. I do apologize for not acknowledging your response earlier. I had already read it a while ago and don't have any follow-up questions. I am satisfied with your response, but I will still keep by my original score of 6.

---

### Official Review · Reviewer_kPYj · 2022-10-22

**Confidence:** 4
**Correctness:** 3
**Technical Novelty And Significance:** 2
**Empirical Novelty And Significance:** 3
**Recommendation:** 6

**Clarity, Quality, Novelty And Reproducibility:**

**Clarity**: overall clear and good writing. Just a couple of minor errors, see minor comments below.

**Quality**: see discussion of strengths & weaknesses above.

**Novelty**: the paper looks good in terms of novelty to me (the authors discuss clearly-related work on similar scaling laws outside of RL, but it is interesting and novel to see something similar for RL).

**Reproducibility**: good, URL with source code provided, and experiments described with sufficient clarity and amount of detail.

**Minor Comments**:
- p. 2: "These Games are selected" --> games should not be capitalised
- p. 2: "In step with scaling observed in the game Hex" --> what does "In step with" mean? Do you mean "In contrast to" or "Similar to"?
- p. 3: "by the MCTS by counting the number of explored future states" --> personally I find this vague, and would simply describe it as the "number of visits" instead of "number of explored future states". I suppose it's usually the same thing (if MCTS expands/explores one state every iteration), but technically visit counts is more correct for end-game situations (where some MCTS iterations might not expand any new state anymore if the tree traversal leads to a terminal state that is already part of the tree, but it does still increment the visit counter).
- p. 9: "but the proof of power-law scaling" --> I'd prefer "evidence" instead of "proof". I prefer to reserve the word "proof" only for cases where something is actually theoretically proven. Empirical experiments just provide evidence, no matter how convincing they may be.
- p. 10: The two "Anonymized" references should have just been provided in a non-anonymized form. They could just have been cited in the third person, as if they were other people's work, without mentioning it was your work. **Do not updated it now during this review cycle though**, or it would reveal your identities.
- p. 15: "Medium-seized" --> Medium-sized

**Question**: When evaluating playing strength of agents, do you still restrict agents by MCTS iteration counts (as in training), or by time limits? I assume that larger neural networks have greater computational overhead, and therefore could produce a weaker full (search+DNN) agent when restricted by time budgets, due to slowing down the search. This could be another reason why, for example, it might have been preferable for AlphaGo Zero / AlphaZero networks to be smaller than the predicted "optimal" sizes according to the scaling laws.

**Strength And Weaknesses:**

**Strengths**:
- Very clear writing.
- Good, extensive experiments (albeit restricted to two games, plus one more in supplementary material).
- Interesting and likely-impactful results and conclusions.

**Weaknesses**:
- I think that a few conclusions / remarks about related work need to be toned down, due to the use of only two games, and many similarities between those two games.

**Detailed Comments**
- On page 2, some motivation is provided for the selection of the 2 games is provided, **stating that they are sufficiently different** with respect to branching factors and game length. I agree that they are quite different in terms of branching factor, but in terms of game lengths they are really quite similar: they are both relatively short (25.4 vs 16.8 according to Table 2). There are many other board games out there with substantially greater durations (e.g., Chess, Shogi, Go, Amazons, Hex, Havannah, ...). These two aspects (branching factor and game duration) are also only two, relatively abstract, aspects. We can think of many other properties of board games: What is the board size? What kind of shape/tiling does the board use? What kinds of win conditions are used? What kinds of movement mechanisms are used? And so on. In almost all of these aspects, the two selected games are very similar. They have similar board sizes (and hence input shapes). They both use tilings of squares (although Pentago does at least have the thing where parts of the board get rotated). They both have line-completion goals (no checkmate or capturing or connection or racing or any other kinds of win conditions). Neither game features movement of pieces (as in Chess/Shogi/Amazons/etc.), only placement of pieces (and indirectly some movement in Pentago due to partial board rotations). Neither game features any capturing/removal of pieces. Both games naturally converge towards a terminal position (because pieces never get removed). **Considering all of these aspects together, I would argue that the two selected games are in fact highly similar, having only a few differences.**
- Given my point above, I feel that some conclusions need to be weakened and limitations more explicitly acknowledge. **Firstly, while I do think the attempt at extrapolating to AlphaZero/AlphaGo Zero is interesting to see, everything related to this needs to be presented with a lot more nuance and acknowledgement of the major differences.** Go is a very different game from Connect Four and Pentago in many important aspects: significantly longer games, much bigger board, very different style of win condition, involves capturing of pieces, does not necessarily naturally converge to a terminal position (I guess it does under expert play when the board gets filled up, but not under non-expert play where giant portions of the board might be freed up during gameplay). The same holds for Chess and Shogi (*are these games actually included when you say "AlphaZero" in e.g. Figure 1?*). **Secondly, in general the main conclusions of the paper about power-law scaling being likely to exist for AlphaZero in general**, even if the exact coefficients may be different, **should again be stated with much more nuance and acknowledgement of the fact that the two evaluated games are extremely similar in many aspects**. Demonstrating this behaviour in two highly-similar board games is not representative of the overall space of board games.
- In principle, I am fine with having used only these 2 games. I understand that computational limitations are an issue, and I also think there are very good reasons to select 2 relatively small games like this (for example because it allows for comparisons against near-perfect players). While more games, bigger games, and more variety in games could've made the work even stronger, I do not see it as a reason to reject. However, I do see it as something that needs to be more explicitly acknowledged as a limitation.

**Summary Of The Paper:**

The authors of this paper ran a large amount of AlphaZero-based training runs, with many different sizes of neural networks, for two different board games: Connect Four and Pentago. Based on these experiments, the paper demonstrates for both games, very similar power laws exist that predict how the playing strength (measured in Elo) scales with respect to various aspects such as compute, num NN params, and generated self-play data.

**Summary Of The Review:**

Overall a good paper, but some of its limitations need to be more explicitly acknowledged, and some conclusions / statements about related work be made with more nuance.

---

> ### Author Response · Authors · 2022-11-15
> **Reply to reviewer kPYj**
>
> Thank you for your positive review, finding our results and conclusions Interesting and likely to be impactful, and seeing our paper as novel and clearly written.
> In particular we are grateful that you acknowledge the difficulties of obtaining the required data.
>
>
>  The only weakness listed regards the language used when making claims about generalization.
>
>  We agree. We realize that our visual comparison to SOTA models combined with the language used gave the appearance that we would claim that the scaling laws found are valid universally for other games. That was not our intention. The relevant parts of the paper were edited accordingly. We now specifically mention potential causes that can invalidate our scaling laws in a new subsection. In addition, it seems this misunderstanding caused some of the other reviewers to assume our main result would be about a general scaling law for all games, which it is not. Our main claim is the evidence of power law scaling laws in some games for AlphaZero, for which we provide significant evidence.
>
> In detail, the changes made are:
> * We removed all claims from the abstract, discussion and figure captions regarding the optimal use of compute in training SOTA models.
> * We added a new subsection (5.4.1) where we discuss the possible problems of extrapolating our results to ResNet agents playing Go, Chess or Shogi, as was done in previous works. We also detail why one may want to train smaller agents in order to fit into inference-time compute constraints.
> * We changed the language of our comparison of Connect Four and Pentago in sections 1 and 4 to reflect their similarities.
> * We also changed wording from 'providing proof' to 'providing evidence' following your suggestion.
>
> We hope this helps clarify our intended claim about the existence of scaling laws.
>
> We will also address the two questions you raised during your review:
>
> > The same holds for Chess and Shogi (are these games actually included when you say "AlphaZero" in e.g. Figure 1?)
>
> Yes, they are. The architecture used for all three games is identical except for the input and output layers, resulting in very small differences in parameter count and forward-pass compute. We initially plotted all three but they could not be visibly distinguished in the plot, so we just labeled it 'AlphaZero'. The same is true for the 3-day-training version of AlphaGo Zero, which differs from AlphaZero in terms of training procedure but not in compute or parameter count. It was omitted from the plot for this reason.
>
> > Question: When evaluating playing strength of agents, do you still restrict agents by MCTS iteration counts (as in training), or by time limits? I assume that larger neural networks have greater computational overhead, and therefore could produce a weaker full (search+DNN) agent when restricted by time budgets, due to slowing down the search. This could be another reason why, for example, it might have been preferable for AlphaGo Zero / AlphaZero networks to be smaller than the predicted "optimal" sizes according to the scaling laws.
>
>  Thank you for this interesting question, which was also raised by commenter David J Wu. We always fix the number of MCTS steps to 300 both during training and evaluation, as mentioned in appendix A. Since we see the importance of measuring performance under a time limit, we ran new experiments and added them in a new appendix section, section C.
>  Plotting model size scaling of Connect Four agents under several time limits, we find an additional power law. The new power law has a smaller scaling exponent than that of fixed MCTS size, which makes sense since by giving a time limit smaller agents receive an advantage over larger agents. We thank you for the insight that led to the exploration of this scaling law, which applies to a more realistic use of AlphaZero.

---

> > ### Comment · Reviewer_kPYj · 2022-12-07
> > **Response to Authors' Response**
> >
> > Apologies, I didn't realise I hadn't acknowledged your response to my review yet. I had already read it a long time ago, and do not really have any follow-up questions and am overall satisfied with your response. I do still stand by my original score of a 6 though (which was already above the acceptance threshold). On a more granular scale I probably would've viewed it as something closer to a 5.5 prior to your revisions, and closer to a 6.5 now, but the form on OpenReview only allows jumping from 6 to 8 (which I feel would be a bigger jump in score than warranted).

---

### Official Review · Reviewer_537d · 2022-10-24

**Confidence:** 4
**Clarity, Quality, Novelty And Reproducibility:** This paper is clearly written.
**Correctness:** 2
**Technical Novelty And Significance:** 2
**Empirical Novelty And Significance:** 2
**Recommendation:** 5

**Strength And Weaknesses:**

# Strength

- This paper is clearly written.
-  Most of the experiments are repeated with several different random seeds, and the results displayed align well with authors' interpretation.


# Weakness

- Since there is neither new algorithm proposed nor insightful phenomenon observed throughout the experiments, I am little bit concerned with the technical novelty of this work.

- It is likely that the discovered scaling law only holds under the very specific training methods, the model architecture and the game environment. Once you changed any of these factors, the scaling law might crash or hold with a very different exponent parameter. The experiments in the current submission cannot rule out such possibility.

- I am also not convinced by the motivation of studying the scaling law for this very specific algorithm in these two moderate-size boardgames. It is unclear to me what insight these results can provide for both practitioners and theorists.

- The authors claimed "This scaling law implies that previously published state-of-the-art game-playing models are significantly smaller than their optimal size, given the respective compute budgets" but to me it is more likely that (1) the previous works might have used different training (engineering) techniques from this paper, and (2) the scaling law discovered in this paper does not generalize to more complex games of different styles (like Go) and even if it can generalize, it may hold with a different exponent parameter. Therefore, I find such claim scientifically misleading.

- The Elo score highly depends on the strength and the playing style of other players. In this paper, other players are generated from AlphaZero with different training configurations. It is unclear to me whether the same scaling law still holds or not if I generate the opponent players in a different way.





**Summary Of The Paper:**

This paper studies the scaling law of AlphaZero algorithm in the MARL setting. Specifically, it uses the Elo rating as the performance criteria, and demonstrates that there is power-law scaling between the Elo rating and the model size (also the computation budget).

**Summary Of The Review:**

So far I am fairly concerned with the novelty and significance of the current submission. I am willing to raise my score if the authors can address my concerns.

---

> ### Author Response · Authors · 2022-11-15
> **Reply to reviewer 537d (1/2)**
>
> Thank you for your review and your positive comments on the clarity of our paper and about our results fitting well with our interpretation.
> We see your main concern is the novelty of our results, and we can gladly explain why we believe they are novel and important, as other reviewers have pointed out too.
> We will go point by point:
>
>
> > Since there is neither new algorithm proposed nor insightful phenomenon observed throughout the experiments, I am little bit concerned with the technical novelty of this work.
>
> We respectfully disagree with the claim that no insightful phenomena are observed in our experiments. The phenomenon our paper describes, neural power law scaling, has received significant attention in the last two years. A notable amount of papers has been published on scaling laws observed in various combinations of model architectures and tasks, in different fields like NLP and vision. These laws have directly led to the creation of several large language models, like GPT-3 [1] and Chinchilla [2]. We cite a small selection of papers dealing with neural scaling laws in references [2-10]. The common element in all those papers is that they describe power law scaling of supervised learning models. So far no analysis of power law scaling laws has been published for a reinforcement learning or a MARL algorithm, although evidence of such scaling went undetected in some cases as we describe in Section 2 of our paper. Our paper is therefore the first extensive analysis of power law neural scaling in RL, making it in our opinion a significant contribution to the field.
>
>
> > It is likely that the discovered scaling law only holds under the very specific training methods, the model architecture and the game environment. Once you changed any of these factors, the scaling law might crash or hold with a very different exponent parameter. The experiments in the current submission cannot rule out such possibility.
>
> You are correct that we cannot rule out the possibility that the scaling laws we found will not fit other specific games and architectures. Validation would have to be done individually for every game in question. We do not see why you find this possibility likely, and would be happy to engage in further discussion if a motivation for this conclusion could be provided. On the contrary, some power law exponents were shown to hold across a wide range of different tasks [5]. In any case, our paper is about the existence of power law scaling laws in AlphaZero when applied to the games we examined, for which we believe we have shown sufficient evidence.
>
>
> > I am also not convinced by the motivation of studying the scaling law for this very specific algorithm in these two moderate-size board games. It is unclear to me what insight these results can provide for both practitioners and theorists.
>
> We believe, and so do reviewers kPYj and T3hx, that the existence of neural power-law scaling laws in RL is significant and impactful for the field. We make this claim given the impact the discovery of neural scaling laws in supervised learning had on the field. Regarding the scope of our work, seminal papers in the field have likewise analyzed single architectures trained on single datasets [2,3] due to the exponential compute costs of correctly fitting neural scaling laws. Training agents on the three games presented in our paper required a considerable amount of time, and performing a wide survey of different algorithms and different games is beyond the scope of this paper. We intend to explore more algorithms and games in future works.
>
>
> > The authors claimed "This scaling law implies that previously published state-of-the-art game-playing models are significantly smaller than their optimal size, given the respective compute budgets" but to me it is more likely that (1) the previous works might have used different training (engineering) techniques from this paper, and (2) the scaling law discovered in this paper does not generalize to more complex games of different styles (like Go) and even if it can generalize, it may hold with a different exponent parameter. Therefore, I find such claim scientifically misleading.
>
> We realize that the language used by us gave you and other reviewers the impression that we claim our results generalize to SOTA models. We apologize for the misunderstanding, since we do not wish to make this claim. In order to clearly state that we agree with your point we edited any mentions of performance of other models on other games, including the one you quote. We also created a new subsection, 5.4.1, detailing the possible caveats in extrapolating our results to SOTA models.
> Regarding your first point, we did in fact use the exact same training algorithm used by Deepmind in [11] (excluding hardware acceleration methods), which is detailed in their paper and is implemented in the OpenSpiel library, created by Deepmind themselves.

---

> > ### Author Response · Authors · 2022-11-15
> > **Reply to reviewer 537d (2/2)**
> >
> >
> > > The Elo score highly depends on the strength and the playing style of other players. In this paper, other players are generated from AlphaZero with different training configurations. It is unclear to me whether the same scaling law still holds or not if I generate the opponent players in a different way.
> >
> > Thank you for bringing this issue to our attention. It is indeed a valid concern given the vulnerability of Elo rating with respect to biased inputs. We therefore ran new experiments and added them in a new appendix section, section C. In the new section we show how calculating Elo only from matches against a new class of noisy-solver agents for Connect Four reproduces our main results, even though they use different playing strategies. We hope our new results answer your concern.
> >
> >
> > We hope that we have provided a satisfying answer to all your concerns. If you do not have further concerns, we hope you might consider raising your score.
> >
> >
> > ### References
> >
> > [1] Brown, Tom, et al. "Language models are few-shot learners." Advances in neural information processing systems 33 (2020): 1877-1901.
> >
> > [2] Hoffmann, Jordan, et al. "Training Compute-Optimal Large Language Models." arXiv preprint arXiv:2203.15556 (2022).
> >
> > [3] Kaplan, Jared, et al. "Scaling laws for neural language models." arXiv preprint arXiv:2001.08361 (2020).
> >
> > [4] Hestness, Joel, et al. "Deep learning scaling is predictable, empirically." arXiv preprint arXiv:1712.00409 (2017).
> >
> > [5] Henighan, Tom, et al. "Scaling laws for autoregressive generative modeling." arXiv preprint arXiv:2010.14701 (2020).
> >
> > [6] Gordon, Mitchell A., Kevin Duh, and Jared Kaplan. "Data and parameter scaling laws for neural machine translation." Proceedings of the 2021 Conference on Empirical Methods in Natural Language Processing. 2021.
> >
> > [7] Zhai, Xiaohua, et al. "Scaling vision transformers." Proceedings of the IEEE/CVF Conference on Computer Vision and Pattern Recognition. 2022.
> >
> > [8] Bello, Irwan, et al. "Revisiting resnets: Improved training and scaling strategies." Advances in Neural Information Processing Systems 34 (2021): 22614-22627.
> >
> > [9] Rosenfeld, Jonathan S., et al. "A constructive prediction of the generalization error across scales." arXiv preprint arXiv:1909.12673 (2019).
> >
> > [10] Sorscher, Ben, et al. "Beyond neural scaling laws: beating power law scaling via data pruning." arXiv preprint arXiv:2206.14486 (2022).
> >
> > [11] Silver, David, et al. "Mastering chess and shogi by self-play with a general reinforcement learning algorithm." arXiv preprint arXiv:1712.01815 (2017).

---

> > > ### Comment · Reviewer_537d · 2022-12-08
> > > **Reply to authors' response**
> > >
> > > Apology for the late reply and thanks for the detailed response. I am overall satisfied with authors' response to my questions, although I am still not that convinced why it is of practical interest to discover such scaling law in RL. I have raised my score from 3 to 5 to acknowledge that my other concerns have been well addressed.
> > >
> > > P.S. I will not argue against accepting this paper, if other reviewers believe its quality is above the bar of ICLR.

---

### Official Review · Reviewer_6fgP · 2022-10-25

**Confidence:** 4
**Correctness:** 2
**Technical Novelty And Significance:** 4
**Empirical Novelty And Significance:** 4
**Recommendation:** 6

**Clarity, Quality, Novelty And Reproducibility:**

Clearly communicated and well written.




**Strength And Weaknesses:**

### Strengths

Interesting results.
Proof of existence is good

### Weaknesses

I think this paper misses the point of the LM scaling laws. Here Kaplan’s results were only significant because we found that the log_loss score for language modeling was very indicative of success for  downstream LM tasks. Here we don’t have this justification for why we should care about ELO.

ELO does relate to performance on the task but i’d expect that to change with compute and model size - the claims for this being significant are only:

* It looks like a scaling law
* This is true for multiple games (sample size 2)

For this result to be compelling I'd either like to be shown over multiple methods (PPO, DQN)  or many many more games.  Simple games such as pong or gridworlds would also be welcomed.

he state space argument really doesn’t make sense to me - I would try discrete the space or form some 1 hot encoding (as done by most PPO implementations for OpenAI gyms' continuous action problems).

Genuinely i can’t tell if this rule generalises to all 2player zero-sum games or a specific niche currently.



**Summary Of The Paper:**

The authors claim to find a relationship between compute, model size and performance (here measured by ELO). Studies are conducted over 2 games (Connect Four and Pentago) and the AlphaZero algorithm. Finally the authors combine these relationships to make recommendations based on optimal model size given the amount of compute available.

The authors also have a third game (Oware) over which they observe a scaling law, which different scaling parameters.


**Summary Of The Review:**


Interesting results but the claim is way too broad for the lack of empirical work done to verify it.

---

> ### Author Response · Authors · 2022-11-15
> **Reply to reviewer 6fgP (1/2)**
>
> Thank you for your positive comments, finding our paper interesting with a good proof of existence, clearly communicated and well written. We are also thankful for your constructive suggestions.
>
> We will address the issues and suggestions raised one by one.
>
>  > I think this paper misses the point of the LM scaling laws. Here Kaplan’s results were only significant because we found that the log_loss score for language modeling was very indicative of success for downstream LM tasks. Here we don’t have this justification for why we should care about ELO.
>
> Finding clear Elo scaling is important since Elo is a direct measure of task performance, as the reviewer pointed out. LM scaling laws of cross-entropy loss are indeed not significant without the connection between test loss and downstream task performance, measured in other quantities such as accuracy. This connection translates cross-entropy loss, which is not necessarily a good measure of success, to well established measures of success like prediction accuracy.
> In contrast, Elo is connected to the probability of winning against any opponent, parallel to accuracy in LMs. Elo is the standard performance measure for both AI and human players in two-player zero-sum games, which is why it is the traditional measure of task performance in MARL. Since our scaling laws scale Elo directly there is no need for an extra step to justify their usefulness.
>
> > ELO does relate to performance on the task but i’d expect that to change with compute and model size
>
> We respectfully disagree with this prediction, since we find no evidence in our results that would support this claim. We would be glad to discuss this further if a justification for assuming that scaling trends break at larger scale would be presented. In general, language and vision scaling laws were shown to be extremely robust across many orders of magnitude, so we see no reason to assume the contrary is true for reinforcement learning.
>
> > For this result to be compelling I'd either like to be shown over multiple methods (PPO, DQN) or many many more games. Simple games such as pong or gridworlds would also be welcomed.
>
>  We definitely agree that looking for scaling laws in other RL architectures and environments is important, and will give more credibility to our results. We intend to do that in future work, but this is beyond the scope of the present paper. Due to the nature of neural scaling laws, obtaining evidence of their existence requires training large models to convergence which can be extremely expensive in compute. Many important papers on this topic are based on scaling laws obtained for a single architecture trained on a single or few datasets. Take for example Kaplan et al. (2020) and Hoffmann et al. (2022), arguably the two most important papers on LLM scaling laws. Kaplan et al. present scaling laws for a single architecture (Transformers) trained on a single dataset (WebText), with the exception of one scaling law for LSTMs in model size. Hoffmann et al. train a single architecture on a single dataset as well, adding an appendix comparing only two models trained on different subsets of this dataset. Hoffmann et al. did indeed measure downstream performance on a multitude of tasks, but they could do that because this is not as computationally expensive as the training phase.
> Doing a similar analysis in RL involves even more difficulties, since training is far less reliable than in supervised learning and a thorough analysis requires training several random seeds for each datapoint, which we did. Obtaining the scaling laws for the three games present in this paper required a significant amount of time. Adding more games is therefore a task for future studies. We also cannot test our trained models on a variety of downstream tasks since, like many other MARL algorithms, AlphaZero is not built for generalizing beyond the task it was trained to do. The only other paper on AlphaZero scaling (Jones, 2021) also explored a single architecture on a single game for similar reasons.
>
> To summarize, we agree that extending this study to a larger number of games and models is important. However we believe that it is beyond the scope of a single paper.
>
> > The state space argument really doesn't make sense to me - I would try discrete the space or form some 1 hot encoding (as done by most PPO implementations for OpenAI gyms' continuous action problems).
>
> We assume you are referring to our hypothesis regarding the difficulty of training Oware agents. We are thankful for your suggestion, with which we agree. One hot encoding and binary board representations are indeed the popular way to implement MARL environments. We are not sure why Deepmind's OpenSpiel uses a different implementation for Oware. Making the suggested changes would require however to change OpenSpiel's source code and to retrain all Oware agents.

---

> > ### Author Response · Authors · 2022-11-15
> > **Reply to reviewer 6fgP (2/2)**
> >
> > > Interesting results but the claim is way too broad for the lack of empirical work done to verify it.
> >
> > We respectfully disagree that the claim is too broad. Our main claim is the existence of power law scaling phenomena in AlphaZero, for which we give a significant amount of evidence. We do not claim all games will produce such scaling laws, although we find it likely enough to be worth further investigation, as we mentioned in the discussion section. We understand that the language we used caused you and other reviewers to view our claims as much broader, which we apologize for. We therefore significantly changed all claims made in the paper about other games and other architectures, adding a new subsection (5.4.1) to specifically detail the caveats of generalizing these scaling laws to other cases.
> >
> > We will be happy to discuss any further concerns you may have. If we addressed all of your concerns, we hope you might consider raising your score.
> >
> > ### References:
> >
> > * Kaplan, Jared, et al. "Scaling laws for neural language models." arXiv preprint arXiv:2001.08361 (2020).
> >
> > * Hoffmann, Jordan, et al. "Training Compute-Optimal Large Language Models." arXiv preprint arXiv:2203.15556 (2022).
> >
> > * Jones, Andy L. "Scaling scaling laws with board games." arXiv preprint arXiv:2104.03113 (2021).

---

> > > ### Comment · Reviewer_6fgP · 2022-12-07
> > > **Response to rebuttal**
> > >
> > > Dear Authors,
> > >
> > > Thank you for your responses. I think having read the revised version of this paper, I'm much more comfortable it being accepted as a documentation of interesting phenomena .
> > >
> > > I will update my score to a 6.

---

### Public Comment · ~David_J_Wu1 · 2022-11-07
**Methodology for the agents being evaluated?**

As a researcher and engineer with experience in computer game playing AI, I would like to say that I find the topic of this submission to be of interest, so I thank the authors for attempting to explore it!

I'm commenting because I'd like to draw additional emphasis to a question from reviewer kPYj: when evaluated for playing strength, were the final agents using a fixed number of MCTS simulation counts per move, or a fixed amount of time per move, or something else? While there are good reasons why a given experiment might use the former (e.g. simulation count limits are hardware-independent, while wall-clock time limits are not), in domains where search is possible to freely scale any model to any inference-time compute budget, optimizing performance per final inference time or cost is usually the practical objective. I and many in my field have all too commonly trained a larger model, found it was much stronger on a fixed-simulation basis, yet gave a worse final agent since the larger cost prevented deeper search in the same time. AlphaZero's model size would likely have been chosen with the same consideration.

Given how hugely the interpretation of some results may hinge on this detail, I found it surprising that only one reviewer also raised this question. I hope it can be easy to clarify, as it would help with interpreting the data presented. Thanks!

---

> ### Public Comment · ~Evgenii_Zheltonozhskii1 · 2022-11-09
> **Evaluation methodology**
>
> My read from the following:
> "The benchmark player is based on the best performing Pentago agent trained, given 10 times more MCTS steps per move."
> is that a fixed number of steps is used. Of course, the authors should have mentioned that explicitly.
> Indeed, using fixed time is a setting more relevant for real life, and wouldn't expect a power law (even saturating one, as in this experiment) there. In addition, being able to use only small networks due to the relative simplicity of the games makes me a bit suspicious regarding the extrapolation.

---

> > ### Author Response · Authors · 2022-11-15
> > **Reply on evaluation methodology**
> >
> > Thank you for showing interest in our paper. We indeed use a fixed number of MCTS steps during matches, as we stated in appendix A.
> > If you look at the new appendix B you might find it interesting that a power law does emerge in this setting, with a different exponent reflecting the compute constraint.

---

> ### Author Response · Authors · 2022-11-15
> **Reply on evaluation**
>
> Thank you for showing interest in our paper and for bringing this to our attention.
> Regarding your question, we use 300 MCTS steps for all agents during both training and matches, as we state in appendix A.
>
> Following your comment, we added a new section to the appendix (section B) where we compare agents under the constraint of fixed inference-time compute.
> It seems an additional power law emerges at the limit of large inference-time compute, with a smaller exponent (makes sense given the advantage small agents receive).
> We hope these results will be useful to the MARL community given its more practical setting.

---

### Author Response · Authors · 2022-11-15
**Revision summary**

We thank all reviewers and public commenters for their constructive feedback. We are glad to see that reviewers find our paper interesting and clearly written, and recognize the importance of our results.

We list here the most important changes we made. We added two new experiments to the paper following questions from reviewers and the public. We believe that the additional scaling law found in one of the additional experiments constitutes a valuable contribution to the paper. We appreciate the suggestions.


### Extrapolation

A main concern by reviewers regards the extrapolation of the optimal-size scaling law we found to the SOTA models AlphaGo Zero and AlphaZero trained on Go, Chess or Shogi. Our original formulation led to the misleading impression that we would expect our scaling laws to describe these models well, which was not our intention. To fix this issue, we introduced several changes to the paper, including:

* Removing all claims from the abstract, discussion and figure captions regarding the optimal use of compute in training SOTA models.
* Adding a new subsection (5.4.1) where we detail the possible problems of extrapolating our results to ResNet agents playing Go, Chess or Shogi, as was done in previous works. We also detail why one may want to train smaller agents in order to fit into inference-time compute constraints.
* Changing our comparison of Connect Four and Pentago in section 4 to reflect their similarities compared to other games.

The paper's main claim is not about extrapolation, but rather that we provide substantial evidence that the AlphaZero algorithm can exhibit power-law scaling with respect to several parameters. All reviewers seem to agree that we provide robust evidence for power-law scaling when AlphaZero is applied to Connect Four and Pentago.


### Inference-time compute scaling

Reviewer kPYj and commenter David J Wu brought to our attention the importance of measuring agent performance under thinking-time limits. Indeed, measuring performance scaling under time limits is highly important for practical purposes. We therefore conducted a series of experiments, which are presented in the new appendix section B.
Plotting Connect Four performance scaling with model size under several different time limits, we find that an additional scaling law emerges at the limit of large inference-time compute. We are thankful for the questions leading to this results, which we think is of great significance to the public, especially for practical implementations of AlphaZero.


### Measuring Elo against other players

Two reviewers raised analogous concerns regarding the reproducibility of our Elo score calculation if we measure agents against players that were not generated with AlphaZero. In order to address those concerns, we ran new experiments where we match our agents against players created by applying a temperature noise to the Connect Four solver. These experiments, presented in the new appendix section C, validate our results by reproducing the scaling exponent we found for Connect Four size scaling. We also elaborate on the vulnerability of Elo rating to biases.


We also removed a part of section 2 to make room for the new additions.

All our code for generating the new experiments has been added to the GitHub repository.

---

### Author Response · Authors · 2022-12-07
**Reviewer responses**

As the discussion period is nearing its end, we would appreciate it if the reviewers could take the time to read our replies and respond to them. We would be glad to answer any concerns remaining after our revision.

---

> ### Author Response · Authors · 2022-12-08
> **Reviewer responses**
>
> We thank all reviewers for their quick response, and for the time they invested in reviewing our paper.

---

### Decision · Program_Chairs · 2023-01-20

**Decision:**

Accept: poster

**Justification For Why Not Higher Score:**

Outstanding concerns from multiple reviewers that findings may not generalize to other games or models/agents.

**Justification For Why Not Lower Score:**

The multiple comments from non-reviewers suggest a broad interest in the topic at ICLR. This could spark discussion between sub-communities at ICLR.

**Metareview: Summary, Strengths And Weaknesses:**

This paper begins to explore how the performance of AlphaZero in two-player competitive games scales in terms of network size. Whilst the reviewers initially raised many concerns, the authors' response led all to conclude willing to accept the paper. The largest outstanding concern was with regard to the generalisation of the results found to other games and models.

Additionally, many reviewers raised interesting future research questions. Out of scope for this initial submission, but showing promising signs this paper could motivate a wide range of further research. Supporting comments from other members of the research community again suggest broad interest in the topic.

**Note From Pc:**

if the above contains the word "oral" or "spotlight" please see: "oral" presentation means -> notable-top-5% and "spotlight" means -> notable-top-25%. As stated in our emails, we are disassociating presentation type from AC recommendations